# Cohort profile: the Oxford Parkinson's Disease Centre Discovery Cohort MRI substudy (OPDC-MRI)

Ludovica Griffanti ,[1,2,3] Johannes C Klein,[1,3,4] Konrad Szewczyk-Krolikowski,[3,4] Ricarda A L Menke,[1] Michal Rolinski ,[3,4,5] Thomas R Barber,[2,3,4] Michael Lawton,[6] Samuel G Evetts,[3,4] Faye Begeti,[3,4] Marie Crabbe,[3,4] Jane Rumbold,[3,4] Richard Wade-Martins,[3,7] Michele T Hu,[3,4] Clare Mackay[2,3,8]

For numbered affiliations see end of article.

**Correspondence to**
Professor Clare Mackay;
clare.mackay@ohba.ox.ac.uk

## ABSTRACT

**Purpose** The Oxford Parkinson's Disease Centre (OPDC) Discovery Cohort MRI substudy (OPDC-MRI) collects high-quality multimodal brain MRI together with deep longitudinal clinical phenotyping in patients with Parkinson's, at-risk individuals and healthy elderly participants. The primary aim is to detect pathological changes in brain structure and function, and develop, together with the clinical data, biomarkers to stratify, predict and chart progression in early-stage Parkinson's and at-risk individuals.

**Participants** Participants are recruited from the OPDC Discovery Cohort, a prospective, longitudinal study. Baseline MRI data are currently available for 290 participants: 119 patients with early idiopathic Parkinson's, 15 Parkinson's patients with pathogenic mutations of the leucine-rich repeat kinase 2 or glucocerebrosidase (GBA) genes, 68 healthy controls and 87 individuals at risk of Parkinson's (asymptomatic carriers of GBA mutation and patients with idiopathic rapid eye movement sleep behaviour disorder-RBD).

**Findings to date** Differences in brain structure in early Parkinson's were found to be subtle, with small changes in the shape of the globus pallidus and evidence of alterations in microstructural integrity in the prefrontal cortex that correlated with performance on executive function tests. Brain function, as assayed with resting fMRI yielded more substantial differences, with basal ganglia connectivity reduced in early Parkinson's and RBD. Imaging of the substantia nigra with the more recent adoption of sequences sensitive to iron and neuromelanin content shows promising results in identifying early signs of Parkinsonian disease.

**Future plans** Ongoing studies include the integration of multimodal MRI measures to improve discrimination power. Follow-up clinical data are now accumulating and will allow us to correlate baseline imaging measures to clinical disease progression. Follow-up MRI scanning started in 2015 and is currently ongoing, providing the opportunity for future longitudinal imaging analyses with parallel clinical phenotyping.

## INTRODUCTION

The Oxford Parkinson's Disease Centre (OPDC)[1] is a multidisciplinary research centre at the University of Oxford supported by Parkinson's UK with funds from The

### Strengths and limitations of this study

► High-quality 3T MRI data in a very well phenotyped and longitudinally followed cohort of Parkinson's and rapid eye movement sleep behaviour disorder were acquired on the same MRI scanner, quite unique for a study of this duration.

► Clinical longitudinal data are acquired every 18 months, information about conversion to Parkinson's of the at-risk individuals will be available, and MRI follow-up is ongoing.

► Statistical maps of published results and support data relative to the analyses are available to share.

► Oxford Parkinson's Disease Centre-MRI phenotyping is deep and relatively frequent, however, the size of the cohort is not at the level of population-level cohort studies.

► MRI sequences are high quality, but could not exploit the latest advances in the field in order to maintain continuity.

Monument Trust. It was established in 2010 and brings together world leaders in clinical neurology, neuroepidemiology, neuroimaging, proteomics, genomics, molecular genetics, transgenic Parkinson's models, neuropharmacology, neurophysiology and neuropathology.

The centre was formed to understand the earliest events in the development of Parkinson's, ultimately with a view to identifying the changes that occur before motor symptoms become apparent.

The overarching goals of the OPDC are to:
► Understand the progression of Parkinson's.
► Predict the onset of Parkinson's.
► Identify potential drug targets for Parkinson's.
► Develop new treatments that will prevent the development of Parkinson's in at-risk individuals.

To these aims, the research activity is structured around three overlapping themes: (1) improved clinical cohorts for development of novel biomarkers; (2) improved cellular and genetic models of Parkinson's pathologies and pathways and (3) novel animal models of early neuronal dysfunction in Parkinson's.

Within theme 1, the OPDC Discovery Cohort is one of the largest and best-characterised cohorts of people with early motor-manifest and prodromal Parkinson's in the world.[2–4] It is a prospective, longitudinal study that has recruited patients with early idiopathic Parkinson's, healthy controls (HCs) and individuals at risk of Parkinson's.

The aim of the OPDC Discovery Cohort is to provide a wealth of data to better understand the biology of premotor and early Parkinson's, and to identify predictors of disease onset and progression. In addition to standardised assessments of motor and non-motor function, there is a particular interest in validating cutting-edge technologies to stratify, predict and chart progression in Parkinson's, including brain imaging, saccadometry, smart-phone and wearable assessments.

The subset of the OPDC Discovery Cohort that underwent brain MRI constitutes the OPDC Discovery MRI (OPDC-MRI) substudy and is the focus of this paper. Driven by the emerging evidence of novel imaging markers with high predictive value, like the study by Vaillancourt et al[5] and others mentioned in,[6] this project started with the collection of a small number of scans. Thanks to the promising results and to accumulating evidence in the field about the potential of MRI to serve as a biomarker for manifest (eg, see refs. [7] [8]) and premotor Parkinson's,[9] the imaging substudy was further expanded, with the aim to develop biomarkers derived from multimodal MRI in order to:

► Detect damage and changes in brain structure and function in early-stage Parkinson's and prodromal 'at-risk' individuals.
► Predict disease progression and understand its neural correlates.
► Stratify at-risk individuals to identify potential candidates for clinical trials.

In this cohort profile, we will describe the cohort composition, the MRI data collected and the processing pipelines that we developed. We will then report findings to date and illustrate our future plans for the cohort.

## COHORT DESCRIPTION
### Eligibility criteria and recruitment
Participants of the OPDC-MRI substudy were recruited from the OPDC Discovery cohort since 2010. For the main study (OPDC Discovery), neurologists, Parkinson's nurses, geriatricians and general practitioners from participating hospitals in the Thames Valley area (total population 2.1 million) were asked to identify all idiopathic Parkinson's cases who were diagnosed by a neurologist (or a geriatrician with a specialist interest in

Parkinson's) within the previous 3 years, according to the UK PD Society Brain Bank Criteria for clinically probable idiopathic Parkinson's disease.[10] Self-selection bias is unavoidable in this scenario. Furthermore, the study group reflects the demographics of the Thames Valley area. All participating clinicians are regularly contacted to ensure screening of incident cases diagnosed since study onset. Eligible cases were approached by post and asked to contact the OPDC if interested in taking part in the study. Patients were assessed in research clinics and their diagnosis was further confirmed by a neurologist specialising in movement disorders. Patients were excluded if confidence in the diagnosis was below 90% at that point. Exclusion criteria for participation assessed by a neurologist specialising in movement disorders are: non-idiopathic parkinsonism, secondary parkinsonism due to head trauma or medication use, cognitive impairment precluding informed consent, dementia preceding motoric Parkinson's by 1-year suggestive of dementia with Lewy bodies, or other features of atypical parkinsonism syndromes such as multiple system atrophy (MSA), progressive supranuclear palsy (PSP), corticobasal degeneration.

More details on the recruitment process, assessment and exclusion criteria for the OPDC Discovery Cohort are described elsewhere.[2] [11] For the MRI substudy, we aimed to scan Parkinson's patients as quickly as possible after enrolment (within 3 years of diagnosis), but due to logistics and patient unavailabilities, scanning was performed up to 6 years after diagnosis (see table 1 for details). Parkinson's patients with more than mild head tremor or presence of dyskinesia/dystonia were excluded since movements artefacts would be likely be too severe to obtain usable images.

Genetic testing for known pathogenic mutation of the glucocerebrosidase gene (GBA; L444P and N370S) and Leucine-rich repeat kinase 2 gene (LRRK2; G2019S and R1441C) was performed on all participants in the OPDC Discovery Cohort who consented to it. In the OPDC-MRI cohort the results are currently available for 82% of the participants for the LRRK2 screening and for 96% of the participants for the GBA screening (two participants did not give consent). The details of the genetic testing procedure are available in reference [4].

The Parkinson's imaging cohort includes 119 sporadic patients (idiopathic Parkinson's disease, iPD), 5 patients with pathogenic mutations of the LRRK2 gene (PD-LRRK2) and 10 patients with pathogenic mutations of the GBA gene (PD-GBA).

The HC group is composed of 68 participants also part of OPDC Discovery. Many of them were spouses and friends of Parkinson's participants with no first-degree or second-degree relatives diagnosed with Parkinson's. HCs were not receiving any medications known to affect the dopaminergic system and absence of Parkinson's diagnosis was confirmed by a neurologist specialising in movement disorder. The at-risk group includes 74 patients with idiopathic rapid eye movement (REM) sleep behaviour

**Table 1** Demographic and clinical characteristics of the OPDC-MRI cohort

|  | iPD | PD-LRRK2 | PD-GBA | RBD | RBD-GBA | aGBA | HC |
|---|---|---|---|---|---|---|---|
| N | 119 | 5 | 10 | 74 | 3 | 8 | 68 |
| Age (years) (mean±SD) | 64.1±10.1 | 66.0±11.6 | 63.8±10.3 | 65.8±7.6 | 61.2±7.4 | 65.9±7.6 | 65.9±8.7 |
| Gender (M/F) | 76/43 | 2/3 | 6/4 | 68/6 | 3/0 | 3/5 | 45/23 |
| Parkinson's disease duration at time of MRI (years) (mean±SD) | (n=117) 2.31±1.52 | 3.06±3.49 | 3.29±1.81 | – | – | – | – |
| Time between MRI and closest clinical assessment (days) (mean±SD) | (n=118) 108±104 | 79±105 | 140±107 | 115±90 | 6±4 | (n=6) 273±25 | (n=64) 387±644 |
| Levodopa equivalent daily dose (mean±SD)* | (n=117) 335±243 | 340±198 | 463±243 | – | – | – | – |
| Hoen and Yahr (mean±SD)* | (n=118) 1.79±0.57 | 2.40±0.89 | 2.10±0.88 | (n=71) 0.03±0.17 | 0 | (n=6) 0 | (n=64) 0 |
| UPDRS III (mean±SD)*† | (n=118) 24.0±10.4 | 41.2±17.8 | 28.4±14.9 | (n=74) 4.5±4.0 | 3.0±1.7 | (n=6) 3.3±4.2 | (n=64) 1.8±2.5 |
| MoCA (mean±SD)*‡ | (n=117) 26.4±2.7 | 26.0±2.2 | 24.4±3.1 | (n=73) 25.6±2.7 | (n=2) 26.0±0.0 | (n=6) 26.7±1.9 | (n=62) 27.5±2.0 |

When data are missing for some participants, the number of values available is specified in brackets.
*Evaluated at closest clinical assessment (controls only have a baseline clinic visit at enrolment).
†Corrected for missing questions using the approach described in Goetz et al.[60]
‡Corrected for education.
aGBA, asymptomatic carriers of a pathogenic mutation of the glucocerebrosidase gene; HC, healthy controls; iPD, idiopathic Parkinson's patients; MoCA, Montreal Cognitive Assessment; OPDC, Oxford Parkinson's Disease Centre; PD-GBA, Parkinson's patients with a pathogenic mutation of the glucocerebrosidase gene; PD-LRRK2, Parkinson's patients with a pathogenic mutation of the Leucine-rich repeat kinase 2 gene; RBD, patients with rapid eye movement sleep behaviour disorder; RBD-GBA, RBD patients with pathogenic mutation of the glucocerebrosidase gene; UPDRS III, Unified Parkinson's Disease Rating Scale - Part III.

disorder (RBD), diagnosed with polysomnography according to International Classification of Sleep Disorders criteria[12] (for more details see Barber et al[4]) patients with RBD with a pathogenic mutation of the GBA gene (RBD-GBA, n=3), and asymptomatic carriers of GBA gene pathogenic mutations (aGBA, n=8).

Additional exclusion criteria for the MRI substudy for all groups were contraindications to MRI scanning, including a history of claustrophobia, incompatible metal foreign body or suspicion of such, unresolved metallic injury to the eye or inability to travel to Oxford without assistance.

### Data collection

Baseline MRI data were collected between November 2010 and December 2018. Follow-up data acquisition started in 2015 and is currently ongoing (see the future plans section). Participants with Parkinson's were scanned in a clinically defined 'off' state, a minimum of 12 hours after the withdrawal of their dopaminergic medications.

### Clinical data

Participants receive extensive assessment in designated research clinics as part of their participation in the OPDC Discovery cohort. The assessment, performed by a nurse and neurologist, includes a structured general medical interview, detailed characterisation of motor and non-motor features, and cognitive assessment (see online supplementary table S1 and [2 13] for details). Patients are followed up clinically every 18 months, while controls only have a baseline clinic visit. In a research clinic, we cannot formally diagnose patients who convert from RBD to Parkinson's (or another neurodegenerative disorder). However, where history and examination suggest conversion to neurodegenerative disease, we alert the treating clinicians who then establish the diagnosis and ensure clinical management is in place. On the day of scanning, an additional Unified Parkinson's Disease Rating Scale Part III (UPDRS III) assessment was performed ('off' in Parkinson's). Table 1 summarises the main demographic and clinical characteristics of the OPDC-MRI substudy cohort.

### Imaging data

Scanning was performed at the Oxford Centre for Clinical Magnetic Resonance Research using a 3T Siemens Trio MRI scanner (Siemens, Erlangen, Germany) equipped with a 12-channel receive-only head coil and foam cushions were used to minimise head motion. The neuroimaging protocol includes both structural and functional sequences and lasts approximately 45–50 min.

Within the allocated time, five modalities were always acquired (*core sequences*), while the remaining time was used to experiment with novel sequences, which changed during the study. We report here the details of two of

**Table 2** MRI core sequences: parameters used in the study and number of available datasets for each modality

|  | T1 | T1 WM nulled | T2-FLAIR | dMRI | rfMRI |
|---|---|---|---|---|---|
| Sequence type | 3D, MPRAGE | 3D, MPRAGE | 2D, FLAIR | EPI | EPI |
| Period of acquisition | 2010–2018 (ongoing) |  |  |  |  |
| TR (ms) | 2040 | 3000 | 9000 | 9300 | 2000 |
| TE (ms) | 4.7 | 3.4 | 90 | 94 | 28 |
| TI (ms) | 900 | 409 | 2500 | -- | -- |
| Flip angle (degrees) | 8 | 8 | 150 | -- | 89 |
| Voxel size (mm$^3$) | 1×1×1 | 0.9×0.9×1 | 1.1×0.9×3 | 2×2×2 | 3×3×3.5 |
| FOV read (mm) | 192 | 240 | 220 | 192 | 192 |
| FOV phase (%) | 90.6 | 81.3 | 100 | 100 | 100 |
| Base resolution | 192 | 256 | 256 | 96 | 64 |
| Phase resolution (%) | 100 | 100 | 75 | 100 | 100 |
| Bandwidth (Hz/Px) | 130 | 130 | 201 | 1628 | 2368 |
| Orientation | Transversal | Transversal | Transversal | Transversal | Transversal |
| N volumes | -- | -- | -- | 60 directions+5 b=0 | 180 |
| Other sequence-specific characteristics | -- | -- | -- | b-value=1000 s/mm$^2$; Echo spacing=0.69 ms | Echo spacing=0.49 ms; Eyes open |
| Acquisition time | 5 m 56 s | 9 m 8 s | 5 m 8 s | 11 m 11 s | 6 m 4 s |
| **No of subjects** |  |  |  |  |  |
| N iPD | 119 | 118 | 100 | 114 | 117 |
| N PD-LRRK2 | 5 | 5 | 1 | 2 | 4 |
| N PD-GBA | 10 | 10 | 1 | 10 | 10 |
| N RBD | 74 | 74 | 68 | 74 | 74 |
| N RBD-GBA | 3 | 3 | 3 | 3 | 3 |
| N aGBA | 8 | 8 | 6 | 8 | 8 |
| N HC | 68 | 66 | 55 | 66 | 68 |
| N total | 287 | 284 | 234 | 277 | 284 |

aGBA, asymptomatic carriers of a pathogenic mutation of the glucocerebrosidase gene; 3D, three dimensional; dMRI, diffusion-weighted MRI; EPI, Echo Planar Imaging; FLAIR, fluid-attenuated inversion recovery; FoV, field of view; HC, healthy controls; iPD, idiopathic Parkinson's patients; MPRAGE, Magnetization Prepared Rapid Acquisition Gradient Echo; PD-GBA, Parkinson's patients with a pathogenic mutation of the glucocerebrosidase gene; PD-LRRK2, Parkinson's patients with a pathogenic mutation of the Leucine-rich repeat kinase 2 gene; RBD, rapid eye movement sleep behaviour disorder; RBD-GBA, RBD patients with a pathogenic mutation of the glucocerebrosidase gene; rfMRI, resting-state functional MRI; TE, echo Time; TI, inversion Time; TR, repetition Time; WM, white matter.

them, introduced at a later stage based on new information becoming available in the field, and which showed promising preliminary results. We describe the other additional sequences we experimented in the online supplementary material.

For each modality, pipelines for preprocessing and analysis of MRI data were developed and applied to the data to extract single-subject imaging variables (both summary and voxel-wise measurements). Here below, we describe for each modality the rationale for acquisition, as well as the main preprocessing steps performed on the images to derive the measures of interest.

The main acquisition parameters and the number of available scans per group for each of the *core sequences* are listed in table 2.

### T1-weighted MRI

The T1w MPRAGE (figure 1A,D) offers very good contrast across tissue classes: grey matter (GM), white matter (WM) and cerebrospinal fluid (CSF). It is primarily used to study GM structural macroscopic tissue in both cortical and subcortical regions. In Parkinson's, cortical morphology in cognitively intact patients is generally reported to be normal or mildly altered, while impaired cognition and dementia in Parkinson's have been found to be associated with more severe patterns of cortical atrophy.[6 14–16] Non-motor symptoms have been also associated with structural changes in specific related brain networks (for a review see ref. 6). With this sequence we aimed to investigate potential differences in GM density across groups or relationships between GM and clinical variables, as well as

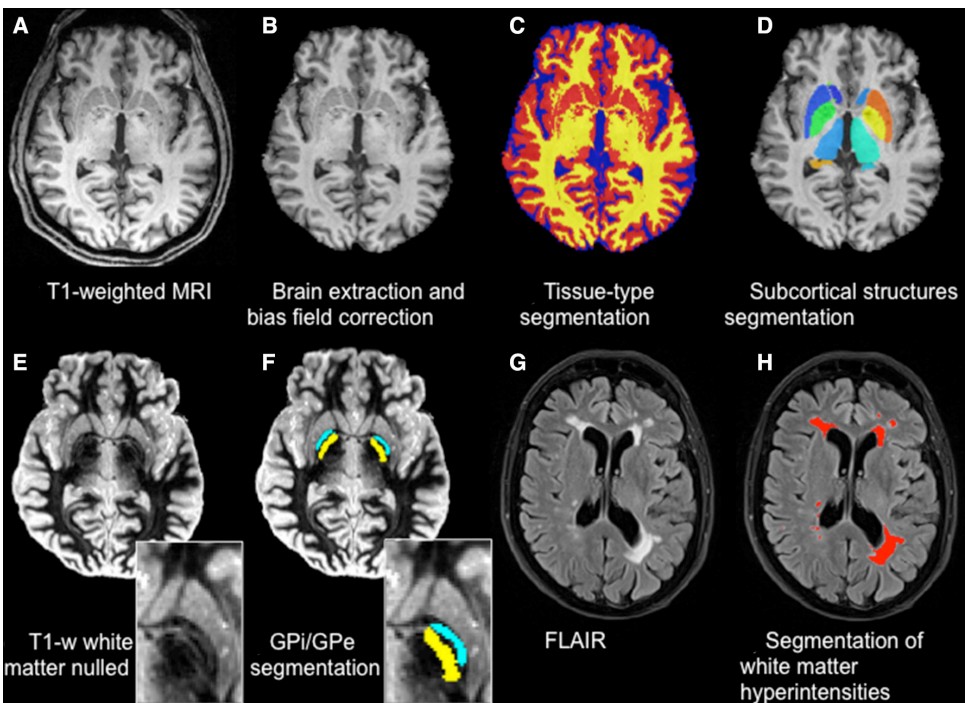

**Figure 1** Structural sequences and related processing. T1-weighted MRI (A) is brain extracted and bias field corrected (B) to perform tissue-type segmentation (GM in red, WM in yellow, CSF in blue) (C) and subcortical structures segmentation (D). T1-weighted white matter nulled (E) allows better contrast in the subcortical structures, which allows, for example, the segmentation of the Globus Pallidus (zoom) into its internal (GPi, yellow) and external (GPe, light blue) portions (F). FLAIR images (G) are used to detect and quantify white matter hyperintensities (red) (H). CSF, cerebrospinal fluid; FLAIR, fluid-attenuated inversion recovery; GM, grey matter; WM, white matter.

taking into account the effect of structural changes when analysing functional MRI data.

The FSL_anat pipeline was used to perform brain-extraction, bias-field correction and 3-class (GM, WM, CSF) tissue-type segmentation with FMRIB's Automated Segmentation Tool (FAST).[17] The resulting images were then used to perform voxel-based morphometry analyses[18] and to generate voxel-wise confound regressors for resting state functional MRI (rfMRI) statistical analyses. Subcortical structure segmentation was performed with FIRST,[19] obtaining three dimensional meshes for each structure for each subject (used to perform vertex analysis), as well as volumetric images used to calculate volumes or as masks to extract average values from other modalities.

T1-weighted images with suppressed WM signal (T1 WM nulled) show enhanced contrast across the basal ganglia structures (figure 1E). This allows the detection of, for example, the boundary between the internal and external globus pallidus and a finer segmentation of this structure using Multimodal Image Segmentation Tool (MIST)[20] (figure 1F).

### T2-weighted fluid-attenuated inversion recovery (FLAIR)

This sequence is commonly used to detect WM abnormalities like leukoencephalopathies, demyelinating diseases and abnormalities of vascular origin. Regarding vascular pathology, it is used to detect WM hyperintensities (WMH) (figure 1G,H). With this sequence, we

wanted to characterise the amount and distribution of WMH in the cohort, look for differences across groups and assess the possible relationship with cardiovascular risk.

FLAIR images were brain extracted and bias field corrected using FAST.[17] WMHs were then automatically segmented on FLAIR images with Brain Intensity AbNormality Classification Algorithm (BIANCA),[21] a supervised segmentation tool which assigns to each voxel a probability of being a lesion, based on their intensity in FLAIR and T1 and their location (details and training dataset are online).[22] The probabilistic output was thresholded and restricted to the voxels located within a mask created from the T1-weighted scans (using the command make_bianca_mask), which excluded the cortex and subcortical structures. The total WMH volume was then calculated for each scan.

### Diffusion-weighted MRI (dMRI)

Diffusion MRI is used to study the microstructure of brain WM in vivo. Also, it serves to virtually reconstruct putative WM tracts. The basis of dMRI is measurement of the random motion of water molecules (diffusion), which has a preferential orientation in the WM (ie, is less restricted along than across the axons). It follows that the preferential direction of water diffusion is related to fibre orientation. The amount of diffusion directionality and restriction can inform about the microstructural environment of a voxel under study.

Diffusion-weighted images were acquired along 60 isotropically distributed diffusion directions (b-value of 1000 s/mm$^2$). Five additional images were acquired without diffusion weighting (b=0 s/mm$^2$). B0 inhomogeneity for diffusion imaging was measured using a dual-echo GRE sequence and the resulting phase and magnitude images were processed to produce field maps for correction of inhomogeneity-induced distortions.

Correction for b0-associated and eddy current-related distortion, as well as participant's movement, were performed using EDDY.[23] EDDY uses a generative probabilistic model to estimate intervolume and intravolume movements, displacements caused by field inhomogeneity, and distortions caused by eddy currents induced by the diffusion gradients. Additionally, automatic artefact rejection replaces slice drop-outs with model estimates.[24 25] The resultant 4D diffusion data were then fed into dtifit, which fits a diffusion tensor model at each voxel[26 27] and generates maps of tensor-derived measures to assess WM microstructural integrity: fractional anisotropy (FA), mean diffusivity (MD) axial diffusivity and radial diffusivity. Finally, we ran bedpostX to produce fibre orientation estimates and their respective uncertainties. This model can be used to perform probabilistic tractography for reconstructing WM pathways and assess their properties such as structural connectivity.

### Resting-state functional MRI (rfMRI)

rfMRI is used to investigate brain function without requiring the subject to undertake a specific task. Although potentially harder to interpret than task fMRI, rfMRI is not affected by subject's performance or compliance and allows to study different resting state networks, that is, sets of brain regions sharing a common time course of spontaneous fluctuations that have been associated with specific brain functions.[28] Among those, the basal ganglia network[29] (shown in yellow in online supplementary figure S1) is of particular interest for the OPDC-MRI cohort. rfMRI data were acquired with eyes open. In 28 iPD patients, we also repeated the rfMRI sequence in the 'on' state 60–90 min after taking their own dopaminergic medication.[11]

First, images were motion corrected with MCFLIRT and the six rigid-body parameter time series extracted for each subject were used for subsequent cleaning. Mean relative displacement was also calculated to potentially exclude subjects with excessive motion and as possible confound metric in further analyses (eg, to verify that there was no significant difference in mean displacement across groups.[30 31] Images were brain extracted, corrected for B0 inhomogeneities using field maps, spatially smoothed using a Gaussian kernel of full width at half maximum (FWHM) of 6 mm, and temporally filtered using a high-pass filtering of 150 s. Single-subject probabilistic independent component analysis (ICA) was then performed with Multivariate Exploratory Linear Optimized Decomposition into Independent Components (MELODIC)[32] with automated dimensionality estimation, followed by automatic component classification with FMRIB's ICA-based Xnoiseifier (FIX)[33 34] to identify the contribution of the artefactual components reflecting non-neuronal fluctuations (FIX training dataset available online).[22] The contributions of the motion parameter time series and of artefactual components were regressed out from the data. The preprocessed functional data were registered to the individual's structural scan and standard space images using FLIRT and FNIRT,[35 36] using boundary-based registration. Single-subject resting state networks were derived with dual regression[37] and compared across subjects. The spatial maps used as spatial regressors in dual regression can be derived either from group-level ICA on the study-specific data[11] or using an external template (for more details regarding this choice see reference 38). Online supplementary figure S1 shows the basal ganglia network map, part of the template used for the analyses. The full templates used for the studies published so far with data from the OPDC cohort are available online.[22]

Regarding non-core sequences acquired in the OPDC-MRI substudy, table 3 shows the main acquisition parameters and the number of available subjects per group for two sequences introduced in 2016, which showed particularly promising results. In the online supplementary material and online supplementary table S2, we provide more information on the other sequences experimented in this study, which include quantitative T1 and T2 mapping, diffusion-weighted imaging of the substantia nigra (SN) and multi-echo T2*-weighted images of the SN.

### Neuromelanin-sensitive MRI (NM-MRI)

The monaminergic neurons in the SN and locus coeruleus (LC) are rich in neuromelanin, a dark pigment that gives these structures their distinct colour. Neuromelanin is detectable with MRI as a hyperintense signal using modified T1-weighted sequences,[39 40] which exploit the paramagnetic properties of the pigment, due to its iron content. According to pathological studies in Parkinson's,[41] the SN and LC are affected early in the neurodegeneration process,[42] making them an interesting target for the development of neuroimaging biomarkers. Studies using NM-MRI found a reduction of the hyperintense signal on NM-MRI in SN and/or LC in patients in patients with established Parkinson's[43 44] and RBD.[45–47] These promising findings drove the inclusion of this sequence in our protocol.

NM-MRI images acquired in OPDC-MRI were bias field corrected using FAST[17] and the transformations to the individual's structural scan (T1w) and standard space were calculated using FLIRT and FNIRT.[35 36] We also defined two reference regions of interest (ROIs) in MNI space (one for SN and one for LC) and used the average intensities within the ROI (registered in individual subject space) as normalisation factors in subsequent analyses.

To also extract quantitative information on these images, we developed a segmentation method to automatically quantify the hyperintense signal in SN and LC

**Table 3** MRI experimental sequences: parameters used in the study and number of available datasets for each modality

| | Neuromelanin-sensitive MRI (NM-MRI) | SWI |
|---|---|---|
| Sequence type | 2D, T1 with MTR† | 3D, T2* weighted |
| Period of acquisition | From 2016, ongoing | |
| TR (ms) | 1400 | 27 |
| TE (ms) | 17 | 20 |
| Flip angle (degrees) | 180 | 15 |
| Voxel size (mm) | 0.8×0.8 x 2 | 0.9×0.9×1.5 |
| FOV read (mm) | 200 | 220 |
| FOV phase (%) | 100 | 90.6 |
| Base resolution | 256 | 256 |
| Phase resolution (%) | 100 | 96 |
| Bandwidth (Hz/Px) | 257 | 120 |
| Orientation | Transversal | Transversal |
| Other sequence-specific characteristics | Reduced FOV‡; GRAPPA (accel factor 2) | Whole brain, GRAPPA (accel factor 2) |
| Acquisition time | 3 m 41 s | 4 m 54 s |
| No of subjects | | |
| N iPD | 28 | 30 |
| N PD-LRRK2 | 1 | 1 |
| N PD-GBA | 0 | 0 |
| N RBD | 45 | 46 |
| N RBD-GBA | 1 | 1 |
| N aGBA | 0 | 0 |
| N HC | 31 | 37 |
| N total | 106 | 115 |

†Twenty-four slices covering the substantia nigra and locus coeruleus. When needed, the number of slices was reduced to remain within the specific absorption rate limits without altering other acquisition parameters.
‡Sequence adapted from Schwarz et al.[43]
aGBA, asymptomatic carriers of a pathogenic mutation of the glucocerebrosidase gene; 3D, three dimensional; FoV, field of View; HC, healthy controls; iPD, idiopathic Parkinson's patients; MTR, Magnetisation Transfer Ratio; PD-GBA, Parkinson's patients with a pathogenic mutation of the glucocerebrosidase gene; PD-LRRK2, Parkinson's patients with a pathogenic mutation of the Leucine-rich repeat kinase 2 gene; RBD, rapid eye movement sleep behaviour disorder; RBD-BGA, RBD patients with a pathogenic mutation of the glucocerebrosidase gene; SWI, susceptibility-weighted imaging; TE, echo Time; TR, repetition Time.

figure 2A,B, preliminary results in reference 48. The analysis on the whole sample is currently ongoing and support data related to the analyses (eg, ROIs) will be available online.[22]

### T2*-weighted images/susceptibility weighted imaging (SWI)

SWI uses tissue magnetic susceptibility differences to enhance contrast in MRI. This is achieved by using the phase image in addition to the magnitude of T2*-weighted images. The phase image contains information about local susceptibility changes between tissues, which can be useful in measuring iron content. There are numerous neurological disorders that can benefit from a sensitive method that monitors the amount of iron in the brain, whether in the form of deoxyhemoglobin, ferritin or hemosiderin.[49] In Parkinson's, SWI has recently emerged as a promising sequence for evaluating the integrity of the SN.[50] In healthy subjects, the dorsolateral SN shows an area of signal hyperintensity, corresponding to nigrosome-I. This feature, described as a 'swallow tail' appearance,[51] is lost in Parkinson's, since nigrosome-I is affected early by synuclein degeneration. Given the promising evidence for the 'swallow tail' to be a candidate biomarker for Parkinson's, we included SWI in our protocol.

The following preprocessing was applied to the T2* images to obtain the final SWI image: macroscopic phase artefacts removal was performed by high-pass filtering the phase images using a 50×50 FWHM window in Fourier space (window size selected empirically to suppress artefacts in the midbrain caused by nearby aerated structures). Then, paramagnetic phase components only were taken to the fourth power and multiplied with the magnitude images.

The presence/absence of the dorsal nigral hyperintensity (DNH), the 'swallow tail' sign, was visually rated for

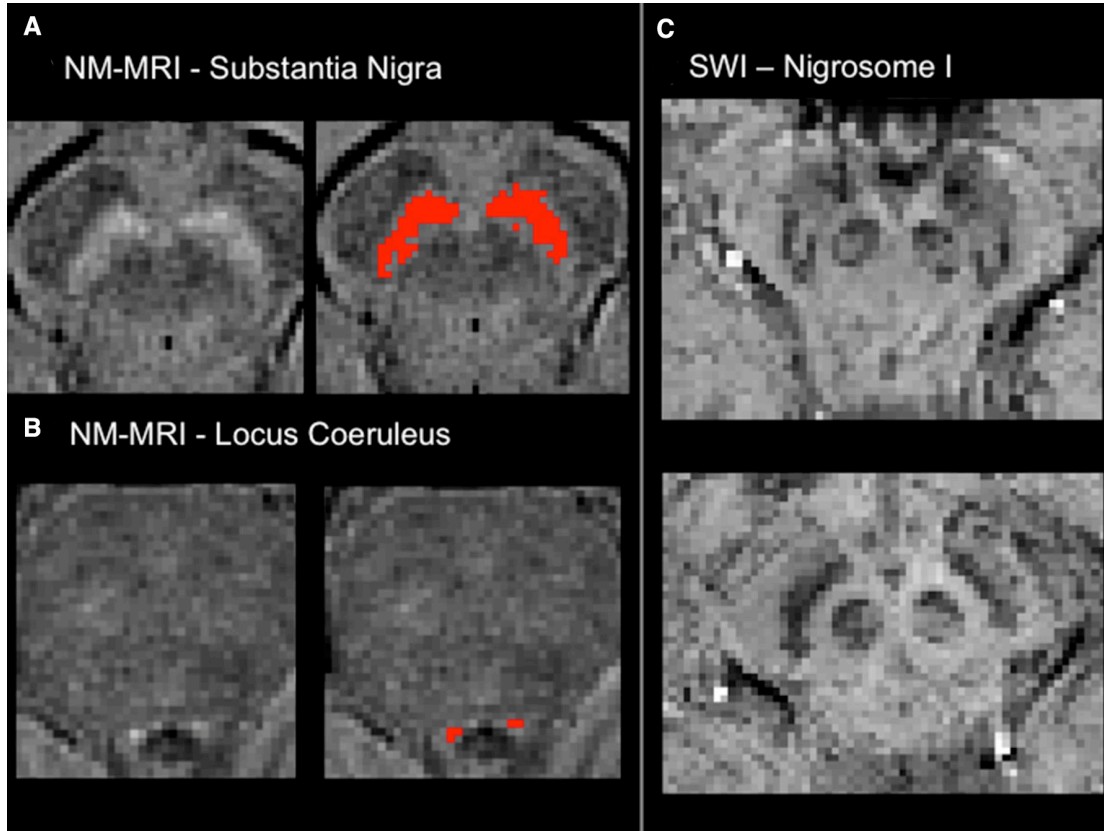

**Figure 2** Novel sequences: Neuromelanin-sensitive MRI (NM-MRI, left) and susceptibility-weighted imaging (SWI, right). Examples of segmentation of (A) the substantia nigra and (B) the locus coeruleus as hyperintense areas on NM-MRI. (C) Examples of the presence (top) absence (bottom) of the 'swallow tail' sign on SWI.

each subject (figure 2C). We also calculated the transformations from the individual's SWI to structural scan (T1w) and standard space using FLIRT and FNIRT[35 36] to perform group comparisons. To extract quantitative information on these images, we also developed a method to automatically quantify the nigrosome-I hyperintense signal.[52] Support data related to the analyses will be available online.[22]

### Patient and public involvement
The OPDC Discovery Cohort is designed by and for patients and is closely linked with the Parkinson's UK local support group. Patient representatives are also involved in the funding/renewal and strategic oversight processes, and sit on the data access panel with casting votes. Results are disseminated to the study participants through annual newsletters, the OPDC website,[1] and series of talks at participants' open days.

### Findings to date
Regarding brain structure, we found no differences between early iPD and controls using relatively standard analyses of whole-brain GM volume and overall regional volumes. A subtle abnormality in the shape of the right pallidus was detected, and corresponded to differences in connecting WM pathways. However, the subtle nature of these changes makes it unlikely that morphometric analysis alone will be useful for early diagnosis of Parkinson's.[53]

We found evidence of alterations in microstructural integrity in the prefrontal cortex that correlate with performance on cognitive tests.[54] This was investigated using cortical measurements of macrostructure and microstructure and performing multimodal linked ICA on structural, quantitative T1 and dMRI in early Parkinson's. Although these patients were cognitively intact, there were significant differences between Parkinson's and controls in a fronto-parietal component primarily driven by altered cortical diffusion (FA and MD). These were related to cognitive performance (Montreal Cognitive Assessment – MoCA). Intriguingly, the frontal areas involved match the distribution of impairment in striatal dopaminergic projections reported previously. A study in an independent patient group found a similar association between frontal WM integrity and cognition, corroborating the idea of early frontal microstructural involvement.[55]

Brain function, as assayed with resting fMRI yielded more substantial differences, with basal ganglia functional connectivity (BGFC) reduced in early iPD and increased on administration of dopaminergic medication.[11] When comparing HC and iPD from the OPDC-MRI substudy with a group of patients with Alzheimer's acquired with the same scanner and protocol,[56] no BGNFC alteration was found in Alzheimer's,[30] providing some confidence that the effect is pathology specific. A similar BGFC

reduction to iPD was found in RBD[31] and was replicated subsequently, although with smaller effect size, in a larger sample of this cohort (https://identifiers.org/neurovault.collection:5686). We did not find significant association between BGFC and motor performance (UPDRS-III)[11 30] also when taking laterality into account.[31] To test whether a link between BGFC and dopamine-related function was present in healthy ageing, we conducted a multivariate analysis in a large population sample of HCs. We found an age-related and sex-dependent decline of connectivity, but no unique dopamine-related function seemed to have a link with BGFC beyond those detectable in and linearly correlated with healthy aging.[57] Our measure of BGFC was reproducible across different analysis settings[38]; however, these group differences have diminished with increasing sample size (online supplementary figure S2) and have not been replicated on a different Parkinson's population so far. At the single-subject level, the discriminatory power of BGFC increased when using a more sophisticated supervised learning algorithm,[58] but further investigation is needed to assess the potential of rfMRI as a clinical biomarker.

Measures of SN and LC volumes extracted from NM-MRI were found to be reduced in patients with RBD compared with controls (preliminary results in reference 48). We also found a decrease in NM in the SN in Parkinson's with respect to controls, especially in Parkinson's with RBD, while the LC seems more affected in RBD, in line with its role in REM sleep regulation.

SWI images showed progressive reduction of the nigrosome-1 signal intensity (DNH—or 'swallow-tail sign') from HC to RBD to manifest Parkinson's in our cross-sectional sample (online supplementary figure S2). In our RBD imaging cohort, 27.5% of patients have pathological nigrosome imaging, defined as absence of the DNH, compared with 7.7% of controls and 96% of patients with Parkinson's. Intriguingly, patients with RBD with DNH absent did not have lower UPDRS-III scores than those with DNH present, but did have reduced dopamine transporter binding in the striatum, suggesting that nigral SWI may be able to identify individuals with dopaminergic decline.[52] This MRI method may have the potential to enrich cohorts for future neuroprotective trials in RBD, where participants with a high likelihood of conversion to motor Parkinson's are sought to achieve clinical endpoints in a manageable timeframe. However, longitudinal follow-up to determine the true predictive value of SWI will be needed to confirm this.

### Long-term follow-up
These baseline data already represent a rich source of data from a deeply phenotyped cohort. A key aspect of the OPDC-MRI cohort, however, is the longitudinal follow-up, which is ongoing. Clinical longitudinal data are acquired in the Discovery cohort every 18 months, and therefore, we will be able to use them to relate baseline imaging with clinical progression. This will potentially allow us to stratify patients and at-risk individuals and

predict their progression. Information about conversion to Parkinson's of at-risk individuals will also be available, providing the ultimate validation of potential biomarkers.

At the latest clinical follow-up visit, two PD participants have been rediagnosed with MSA, three with PSP and one with motor neuron disease. For the remaining patients with PD, the diagnosis was confirmed with diagnostic confidence over 90%, except one with 80% and one with 55%. Among RBD participants, 11 converted after receiving baseline MRI: 6 converted to PD, 2 to dementia with Lewy bodies, 2 to MSA and 1 to pure autonomic failure.

In 2015, we also commenced the acquisition of longitudinal follow-up MRI after 5 years from baseline for Parkinson's and HC, and after 2.5–3 years for RBD. This difference in time after baseline was chosen because we predict that a number of the patients with RBD will convert to Parkinson's, so we aim to collect multiple data points prior to conversion. In this way, we hope to assess their trajectory during the prodromal phase and capture the associated brain changes. So far (December 2019) we have collected 91 follow-up scans: 47 iPD, 2 PD-LRRK2, 19 RBD, 2 RBD-GBA and 21 HC.

### Strengths and limitations of this study
The main strength of this cohort is the collection of high-quality 3T MRI data in a very well-phenotyped longitudinal cohort of Parkinson's and at-risk individuals. The RBD dataset is one of the biggest brain MRI datasets available for this population of at-risk individuals.

Inevitably, a limitation is due to the trade-off between depth of the phenotyping and size of the cohort. OPDC-MRI phenotyping is deep and longitudinal, however, the size of the cohort is relatively modest, particularly when compared with population-level cohort studies.

We are also aware of the inevitable selection bias for this cohort. We recruited people already participating in OPDC discovery cohort, and therefore, our cohort reflects the demographic of the catchment area. Moreover, our participants are those who were already participating in a research study and declared they were happy to be contacted for imaging (self-selection). The exclusion of participants who have severe motor dysfunction, more than mild head tremor or presence of dyskinesia/dystonia, means that the study includes only patients that were able to travel and tolerate an MRI scan 'off' medication. As a result, our cohort cannot fully capture the variability of Parkinson's and RBD patients (eg, patients with severe motor dysfunction may be less able to travel and tolerate an MRI scan; patients with comorbidities such as lung disease may find it hard to lie flat for the scan; severely obese patients cannot be scanned in our system; very apathetic patients may not respond to postal invites).

Another key strength of the OPDC-MRI cohort is the imaging protocol. All MRI data were acquired using the same MRI scanner, which is quite unique for a study of this duration. In this way all baseline data, as well as

longitudinal MRI data will be truly comparable, with no effect due to scanner and/or protocol change. The protocol includes both standard sequences, which are also comparable with other studies, as well as more experimental sequences acquired to investigate study-specific research questions.

We incorporated a trade-off between exploiting the latest techniques available in MRI and continuity of protocol throughout the study. While we could not use the most up-to-date sequences (eg, our EPI images—dMRI and rfMRI—are not acquired with multiband acceleration), we managed to reach a balance by fixing the core sequences to maintain continuity, while changing the experimental sequences as the field evolved to include promising techniques as they became available.

As detailed in the previous section, the cohort will be enriched by collecting longitudinal follow-up MRI data. The challenge will be to be able to get data from those patients who have progressed quickly and will be in a more severe Parkinson's stage. While they may still be able to undergo a telephone interview or clinical assessment, they may not be willing or able to tolerate an MRI scan.

As described in more details in the 'Collaboration and data sharing' section below, we have made statistical maps of our results publicly available, as well as support data relative to the analyses. The data presented here are also available to request (details below).

## Collaboration and data sharing

Details about collaborating with OPDC can be found at https://www.opdc.ox.ac.uk/external-collaborations; OPDC is part of the CENTRE-PD twinning project (https://www.centre-pd.lu), we have ongoing international collaborations and are open for new proposals.

The data presented in this work (baseline imaging, demographics and clinical variables) will be available through the Dementias Platform UK (https://portal.dementiasplatform.uk), where data can be accessed by submitting a study proposal. Please note that longitudinal data will become available at a later stage.

Statistical maps are available on NeuroVault[59] for the following publications:

https://neurovault.org/collections/2694/.[11]

https://registry.identifiers.org/registry/neurovault.collection:5448.[30]

(Results relative to a replication of the original study on an increased sample – see also online supplementary file 1): https://identifiers.org/neurovault.collection:5686.[31]

https://registry.identifiers.org/registry/neurovault.collection:2953.[38]

https://registry.identifiers.org/registry/neurovault.collection:2681.[57]

Other types of support data related to the analyses are available online (https://ora.ox.ac.uk/objects/uuid:8200af66-f438-4a7b-ad14-e8b032f0a9e7)[22] and the repository will keep being populated as the analyses progress.

**Author affiliations**
[1]Wellcome Centre for Integrative Neuroimaging, Oxford Centre for Functional MRI of the Brain, Nuffield Department of Clinical Neurosciences, University of Oxford, Oxford, UK
[2]Wellcome Centre for Integrative Neuroimaging, Oxford Centre for Human Brain Activity, Department of Psychiatry, University of Oxford, Oxford, UK
[3]Oxford Parkinson's Disease Centre, University of Oxford, Oxford, UK
[4]Nuffield Department of Clinical Neurosciences, University of Oxford, Oxford, UK
[5]Institute of Clinical Neurosciences, University of Bristol, Bristol, UK
[6]Population Health Sciences, University of Bristol, Bristol, UK
[7]Department of Physiology, Anatomy and Genetics, University of Oxford, Oxford, UK
[8]Oxford Health, NHS Foundation Trust, Oxford, UK

**Acknowledgements** The authors would like to acknowledge all the participants and their families for participating in the study. They also thank Tim Quinnell, Oliver Bandmann, Gary Dennis, Zenobia Zaiwalla and Graham Lennox for patients recruitment and in-clinic data collection. They are thankful to the staff of the Oxford Centre for Magnetic Resonance (OCMR), in particular Jane Francis, Kathryn Lacey, Rebecca Mills and Steven Knight, and to Amandine Louvel and Katie Ahmed for administering the cohort.

**Contributors** LG had a major role in data acquisition and analysis, interpreted the data and drafted the manuscript for intellectual content. JCK had a major role in data acquisition and analysis, interpreted the data and contributed to major revisions of the manuscript for intellectual content. KS-K, RALM, MR and TRB had a major role in data acquisition and analysis, interpreted the data and revised the manuscript for intellectual content. ML, SGE, FB had a major role in data analysis and revised the manuscript for intellectual content. MC and JR had a major role in data acquisition and revised the manuscript for intellectual content. RW-M, MTH and CM designed and conceptualised the study, interpreted the data and revised the manuscript for intellectual content. All authors reviewed, critically revised and approved the manuscript.

**Funding** The work was supported by the Monument Trust Discovery Award from Parkinson's UK (J-1403) and by the Wellcome Centre for Integrative Neuroimaging, the MRC Dementias Platform UK (MR/L023784/2), the National Institute for Health Research (NIHR) Oxford Biomedical Research Centre (BRC), and the NIHR Oxford Health BRC (a partnership between Oxford Health NHS Foundation Trust and the University of Oxford). JCK acknowledges support from the NIHR Oxford Health Clinical Research Facility. MR received funding support from an NIHR Academic Clinical Lectureship and a NIHR Oxford BRC Doctoral Training Fellowship. TRB received funding support from a Wellcome Trust Doctoral Training Fellowship, and a Biomedical Research Council Career Development Fellowship.

**Disclaimer** The views expressed are those of the authors and not necessarily those of the NHS, the NIHR or the Department of Health.

**Competing interests** MTH reports grants from Parkinson's UK Monument Discovery Award during the conduct of the study; other from Biogen Digital and Roche Prodromal Advisory Boards, outside the submitted work. Other authors have nothing to declare.

**Patient and public involvement** Patients and/or the public were involved in the design, or conduct, or reporting, or dissemination plans of this research. Refer to the Methods section for further details.

**Patient consent for publication** Not required.

**Ethics approval** The study was undertaken with the understanding and written consent of each subject, with the approval of the local NHS ethics committee (National Research Ethics Service (NRES) Committee South Central – Oxford A (Ref:16/SC/0108) and Oxford C (Ref:15/SC/0117); Berkshire Research Ethics Committee (Ref: 10/H0505/71)), and in compliance with national legislation and the Declaration of Helsinki.

**Provenance and peer review** Not commissioned; externally peer reviewed.

**Data availability statement** The data presented in this work will be available through the Dementias Platform UK (https://portal.dementiasplatform.uk), where data can be accessed by submitting a study proposal. Statistical maps are available on NeuroVault.org and support data related to the analyses are available online (https://ora.ox.ac.uk/objects/uuid:8200af66-f438-4a7b-ad14-e8b032f0a9e7).

**ORCID iDs**
Ludovica Griffanti http://orcid.org/0000-0002-0540-9353
Michal Rolinski http://orcid.org/0000-0003-1191-7060

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
