## [Reviewer comments · BMJ Open]

ARTICLE DETAILS

TITLE (PROVISIONAL)	Cohort Profile: the Oxford Parkinson's Disease Centre Discovery Cohort Magnetic Resonance Imaging sub-study (OPDC-MRI)
AUTHORS	Griffanti, Ludovica; Klein, Johannes; Szewczyk-Krolikowski, Konrad; Menke, Ricarda; Rolinski, Michal; Barber, Thomas; Lawton, Michael; Evetts, Samuel; Begeti, Faye; Crabbe, Marie; Rumbold, Jane; Wade-Martins, Richard; Hu, Michele; Mackay, Clare

VERSION 1 – REVIEW

REVIEWER	Sofia Reimão Centro Hospitalar Lisboa Norte, EP - Lisbon, Portugal
REVIEW RETURNED	03-Nov-2019

GENERAL COMMENTS	Griffanti, Ludovica and colleagues presented the characteristics of a Database aimed at gathering high quality multimodal brain MR imaging in association with clinical data of Parkinson's disease (PD) patients and at-risk individuals. This database can be very relevant for PD investigation but the authors, in this paper, present only the cohort profile and very limited preliminary data. All the presented preliminary data needs further work in separate individual publications. Concerning the presented information there are also major issues that need to be addressed. 1. A major issue is concerned with Parkinson's disease diagnosis, a key point in all the gathered data. The authors say that "...all idiopathic Parkinson's cases who were diagnosed by a neurologist (or a geriatrician with a specialist interest in Parkinson's) within the previous three years, according to the UK PD Society Brain Bank Criteria for clinically probable idiopathic Parkinson's disease" (p.5). As clearly demonstrated by previous evidence the diagnostic accuracy of movement disorders is correlated with the training of the physician. Using only clinical criteria for the diagnosis and lacking pathological data, in a project of this magnitude the subject's evaluation needed the expertise of a Neurologist specialized in movement disorders.2. Were healthy control also evaluated by a movement disorders specialist to exclude PD? This is a key issue.3. Also, concerning exclusion criteria, the authors will exclude subjects with "dementia preceding motoric Parkinson's by one year suggestive of dementia with Lewy bodies, or other features of
--

	atypical parkinsonism syndromes such as multiple system atrophy, progressive supranuclear palsy, corticobasal degeneration". How will this evaluation be made? Who will evaluate these subjects? There is a clear need for a Neurologist with movement disorders specialization for an accurate evaluation. 4. The authors will exclude "Parkinson's patients with more than mild head tremor or presence of dyskinesia/dystonia were excluded" (pag 7). This is an important bias in this database that will compromise the results. Patients should only be excluded if the movement artifacts would not allow imaging. 5. It is not clear from the project if all the subjects will be genetically evaluated. If not, what will be the selection criteria for the genetical testing? How will the selection be done? 6. Although the imaging protocol is detailed there is very limited information on the clinical evaluation of the subjects. How will they be clinically evaluated? 7. There is also the need of more information concerning the recruitment for the inclusion in the database. As the authors acknowledge there is a clear selection bias for this cohort, as the participants were already participating in a research study.
--	--

REVIEWER	Roxana Burciu University of Delaware
REVIEW RETURNED	06-Dec-2019

GENERAL COMMENTS	The paper describes The Oxford Parkinson's Disease Centre Discovery Cohort MRI (OPDC-MRI) substudy which has been collecting multimodal MRI data from patients with Parkinson's disease (PD), healthy controls and individuals at risk for PD such as those with RBD since 2010. One unique feature of this database is the fact that all MRI scans were collected on the same scanner and it includes novel MRI sequences including some which have been shown to be sensitive to disease progression in PD (DTI, NM-MRI). The paper discusses brain changes in PD and RBD based on data collected as part of this project and includes some preliminary data on new sequences such as neuromelanin (especially in the RBD cohort ; - a figure would be useful here). Overall, the paper is well-written and the database itself may help better understand the progression of PD as well as brain changes in a subgroup at risk.  • Abstract:  - It includes a reference to AD patients even though they are not included in the section describing participants. Please remove or clarify whether PD were directly compared to AD. Otherwise, interpret results with caution. • Participants:  - Collection of MRI data started in 2015 and follow-up in 2015. It seems that by now there would be sufficient clinical data collected in these groups to run an analysis that looks at whether baseline imaging metrics predict the progression on clinical measures. Please include a description of how many follow-up scans have been collected so far (perhaps include in a table or modify an existing table).
---

	 - The paper would have benefited from preliminary analyses of the baseline imaging data vs. baseline clinical data (UPDRS/MOCA). - Can you comment on the stability of the diagnosis at follow-up? Any statistics on drop-outs or exclusion because of conversion to atypical forms of parkinsonism? Any other neurological disorder besides PD or any concurrent movement disorder such as dystonia? - Were there any behavioral measures collected in addition to the gold standard UPDRS? Any planned analyses? • Resting-state fMRI:  - Please clarify how the data were collected: eyes open (fixation cross/no fixation) vs. eyes closed. - I agree with the authors that resting state fMRI is not affected by the subject's performance and this makes the technique optimal for patients who may have cognitive problems and struggle with task instructions. However, it is well known that in PD this technique gives variable results with many previous studies struggling with validating the results. A major issue here is the fact that a key symptom in PD is tremor at rest (sometimes unilateral, sometimes bilateral) and this likely contaminates the activity. Do the authors control for that in any way? EMG activity during resting state? Rest tremor score from UPDRS as a covariate in the analysis? If not, this should be addressed in the discussion. - The authors used an ICA approach implemented in FSL along with a dual regression to analyze the data. The supplementary material contains an example of the mean resting state activity within the basal ganglia network (defined in MELODIC). It is not clear if the statistical map is that of PD or controls. It is mentioned that basal ganglia functional connectivity was reduced in early PD. Connectivity with what region? It would help to plot that (results of dual regression analysis) rather than the BG network alone. - P value? Correction for multiple comparisons. Please include details on this especially when describing figures (e.g. BG network).
--	--

VERSION 1 – AUTHOR RESPONSE

Reviewer: 1 - Reviewer Name: Sofia Reimão

Institution and Country: Centro Hospitalar Lisboa Norte, EP - Lisbon, Portugal

Griffanti, Ludovica and colleagues presented the characteristics of a Database aimed at gathering high quality multimodal brain MR imaging in association with clinical data of Parkinson's disease (PD) patients and at-risk individuals.

This database can be very relevant for PD investigation but the authors, in this paper, present only the cohort profile and very limited preliminary data.

All the presented preliminary data needs further work in separate individual publications.

As the reviewer correctly points out, there is much work to be done on the data generated which we believe will lead to a stream of separate publications on specific research questions. The paper presented here aims at describing the cohort, with a focus on the demographic composition, imaging protocol, and detailed pre-processing and analysis steps performed. Unfortunately, these details have to be abridged in most research papers due to word constraints, but are key to ensure reproducibility of the results. Research papers that this applies to are summarised in the “Findings to date” section. We are pleased to report that since the first submission of this manuscript, we have published the final results of the analyses on the SWI data, and we have added a summary and relative citation to the “Findings to date” section. Moreover, also in response to the second reviewer, we have now added a supplementary figure (Figure S2) showing some results from the SWI and NM-MRI data as an illustrative example.

Concerning the presented information there are also major issues that need to be addressed.

1. A major issue is concerned with Parkinson’s disease diagnosis, a key point in all the gathered data. The authors say that “...all idiopathic Parkinson’s cases who were diagnosed by a neurologist (or a geriatrician with a specialist interest in Parkinson’s) within the previous three years, according to the UK PD Society Brain Bank Criteria for clinically probable idiopathic Parkinson’s disease” (p.5). As clearly demonstrated by previous evidence the diagnostic accuracy of movement disorders is correlated with the training of the physician. Using only clinical criteria for the diagnosis and lacking pathological data, in a project of this magnitude the subject’s evaluation needed the expertise of a Neurologist specialized in movement disorders.

We can confirm that all eligible cases identified were assessed in research clinics and the diagnosis was confirmed by a neurologist specialising in movement disorder (Prof. Michele Hu and her team) before being enrolled in the study. We have now changed the text accordingly to reflect this:

(Page 5) “... *Eligible cases were approached by post and asked to contact the OPDC if interested in taking part in the study. Patients were assessed in research clinics and their diagnosis was further confirmed by a neurologist specialising in movement disorders. Patients were excluded if confidence in the diagnosis was below 90% at that point.*”

2. Were healthy control also evaluated by a movement disorders specialist to exclude PD? This is a key issue.

Similarly to the patients, we can confirm that the controls were also assessed in research clinics and the PD diagnosis was excluded by a neurologist specialising in movement disorder. The relevant text now reads:

(Page 6) “*The healthy control group is comprised of 68 participants also part of OPDC Discovery. Many of them were spouses and friends of Parkinson’s participants with no first- or second-degree relatives diagnosed with Parkinson’s. Healthy controls were not receiving any medications known to affect the dopaminergic system and absence of Parkinson’s diagnosis was confirmed by a neurologist specialising in movement disorder.*”

3. Also, concerning exclusion criteria, the authors will exclude subjects with “dementia preceding motoric Parkinson’s by one year suggestive of dementia with Lewy bodies, or other features of atypical parkinsonism syndromes such as multiple system atrophy, progressive supranuclear palsy, corticobasal degeneration”.

How will this evaluation be made? Who will evaluate these subjects? There is a clear need for a Neurologist with movement disorders specialization for an accurate evaluation.

Also in this case the text was amended to clearly state that neurologists specialising in movement disorders were involved throughout the study.

(Page 5) *“Exclusion criteria for participation, assessed by a neurologist specialising in movement disorders, are:...”*

4. The authors will exclude “Parkinson’s patients with more than mild head tremor or presence of dyskinesia/dystonia were excluded” (pag 7). This is an important bias in this database that will compromise the results. Patients should only be excluded if the movement artifacts would not allow imaging.

We apologise for the lack of clarity regarding this exclusion criterion. As the reviewer correctly points out, the reason to exclude this type of patients is precisely because movements artefacts would be too severe to obtain usable images. To avoid wasting scanning resources and participants’ time we adopted this clinical criterion as a proxy to judge the likelihood to obtain usable images. We have now rephrased the text as follows:

(Page 6) *“Parkinson’s patients with more than mild head tremor or presence of dyskinesia/dystonia were excluded since movements artefacts would likely be too severe to obtain usable images.”*

We also agree that this creates a bias in the dataset, which we now explicitly acknowledge in the text as follows:

(Page 16) *“Moreover, our participants are those who were already participating in the OPDC Discovery study and declared they were happy to be contacted for imaging (self-selection). The exclusion of participants who have severe motor dysfunction, more than mild head tremor or presence of dyskinesia/dystonia, means that the study includes only patients that were able to travel and tolerate an MRI scan “off” medication. As a result, our cohort cannot fully capture the variability of Parkinson’s and RBD patients...”*

5. It is not clear from the project if all the subjects will be genetically evaluated. If not, what will be the selection criteria for the genetical testing? How will the selection be done?

As part of the main OPDC Discovery cohort, genetic tests are performed on samples from all participants who gave consent. We have now acknowledged this in the text as follows:

(Page 6) “Genetic testing for known pathogenic mutation of the glucocerebrosidase gene (GBA; L444P and N370S) and Leucine-rich repeat kinase 2 gene (LRRK2; G2019S and R1441C) was performed on samples from all consenting participants in the OPDC Discovery Cohort. In the OPDC-MRI cohort the results are currently available for 82% of the participants for the LRRK2 screening and for 96% of the participants for the GBA screening (two participants did not give consent). The details of the genetic testing procedure are available in Barber et al., 2017.”

Details about the genetic testing were not repeated here due to space constraints, but for the reviewer's convenience here is the relevant paragraph from the cited paper:

“Participants were screened for G2019S and R1441C mutations in the LRRK2 gene and N370S and L444P mutations in the GBA gene. [...] DNA was extracted from whole blood using a Qiagen Autopure automated system. Polymerase chain reaction (PCR) was performed using MegaMix Blue (Microzone) containing a recombinant *Taq* polymerase. Primer sequences were as follows: G2019S: 5'-TTTAAGGGACAAA GTGAGCAC-3' and 5'-ACTCTGTTTTCTTTTGAATC-3'; R1441C: 5'-AAGGCATGAAGATGGGAAAG-3' and 5'-TGA TGGTTTTCCGAAGTTTTG-3'; N370S: 5'-GCCTTTGTCCTT ACCCTC*G-3' and 5'-GACAAAGTTACGCACCCAA-3'; L444P: 5'-GGAGGACCCAATTGGGTGCGT-3' and 5'-ACG CTGTCTTCAGCCCACTTC-3' (* indicates a mismatch that was introduced into the forward primer to create a restriction site). The PCR products for G2019S, R1441C, N370S, and L444P were digested with *Sfci* (*Bfml*), *BstUI*, *XhoI*, and *NciI* (*Bcni*), respectively and resolved by agarose gel electrophoresis.”

6. Although the imaging protocol is detailed there is very limited information on the clinical evaluation of the subjects. How will they be clinically evaluated?

Thank you for pointing this out. Assessments of patients were via self-completed questionnaires and from research visits, using standardised and validated scales. Due to word limit constraints and the focus on the imaging data, we decided to add a supplementary table with these details and refer to it and the relevant publications in the main manuscript. However, if the reviewers and the editor think the table should be moved into the main text, we would be more than happy to do so.

The text now reads (Page 7):

Participants receive extensive assessment in designated research clinics as part of their participation in the OPDC Discovery cohort. The assessment, performed by a nurse and neurologist, includes a structured general medical interview, detailed characterisation of motor and non-motor features, and

cognitive assessment (see Supplementary table S1, Szewczyk-Krolikowski et al., 2013 and Rolinski et al., 2014 for details).

7. There is also the need of more information concerning the recruitment for the inclusion in the database. As the authors acknowledge there is a clear selection bias for this cohort, as the participants were already participating in a research study.

We acknowledge that this study, like others, is susceptible to selection bias. For example, the catchment population is overwhelmingly Caucasian, and given that participation is voluntary, self-selection bias is another obvious factor limiting the ability to represent the full spectrum of disease. This will in particular affect recruitment of patients with comorbidities and severe motor disease who may not find it practical to come to research clinic. All participants taking part in this imaging study were already participating in the main research study, the OPDC Discovery cohort. The selection bias therefore percolates into the imaging study presented here. The recruitment process for the main study (OPDC Discovery cohort) is the one we describe in the first part of the “Eligibility criteria and recruitment” section, which is also reported in the two clinical papers cited (Szewczyk-Krolikowski et al., 2013; Rolinski et al., 2014). The OPDC-MRI sub-study only applies some specific MRI-related restrictions to the inclusion criteria for OPDC Discovery, which we detailed in the second part of the section.

We have now better clarified the relationship between the main study and the imaging study and relevant recruitment process, both in the “Eligibility criteria and recruitment” section, as well as in the “Strengths and limitations of this study” section. The relevant text now reads:

Eligibility criteria and recruitment (Page 5):

“Participants of the OPDC-MRI sub-study were recruited from the OPDC Discovery cohort since 2010. For the main study (OPDC Discovery), neurologists, Parkinson’s nurses,...”

...“according to the UK PD Society Brain Bank Criteria for clinically probable idiopathic Parkinson’s disease (Hughes, 1992). Self-selection bias is unavoidable in this scenario. Furthermore, the study group reflects the demographics of the Thames valley area.”

“The healthy control group is comprised of 68 participants also part of OPDC Discovery. Many of them were spouses and friends of Parkinson’s participants with no first-degree relatives diagnosed with Parkinson’s.”

Strengths and limitations of this study (Page 16)

“We are also aware of the inevitable selection bias for this cohort. We recruited people already participating in OPDC Discovery cohort and therefore our cohort reflects the demographic of the catchment area. Moreover, our participants are those who were already participating in a research study and declared they were happy to be contacted for imaging (self-selection). The exclusion of participants

who have severe motor dysfunction, more than mild head tremor or presence of dyskinesia/dystonia, means that the study includes only patients that were able to travel and tolerate an MRI scan “off” medication. As a result, our cohort cannot fully capture the variability of Parkinson’s and RBD patients (e.g. patients with severe motor dysfunction may be less able to travel and tolerate an MRI scan; patients with comorbidities such as lung disease may find it hard to lie flat for the scan; severely obese patients cannot be scanned in our system; very apathetic patients may not respond to postal invites).”

Reviewer: 2 - Roxana Burciu

Institution and Country: University of Delaware

The paper describes The Oxford Parkinson’s Disease Centre Discovery Cohort MRI (OPDC-MRI) substudy which has been collecting multimodal MRI data from patients with Parkinson’s disease (PD), healthy controls and individuals at risk for PD such as those with RBD since 2010. One unique feature of this database is the fact that all MRI scans were collected on the same scanner and it includes novel MRI sequences including some which have been shown to be sensitive to disease progression in PD (DTI, NM-MRI). The paper discusses brain changes in PD and RBD based on data collected as part of this project and includes some preliminary data on new sequences such as neuromelanin (especially in the RBD cohort ; - a figure would be useful here). Overall, the paper is well-written and the database itself may help better understand the progression of PD as well as brain changes in a subgroup at risk.

We thank the reviewer for the positive comments. Also in reply to the first reviewer, we have now added a supplementary figure (Figure S2) showing group average figures for SWI and neuromelanin for the three main groups of the cohort: controls, RBD and PD participants.

Figure S2. Novel sequences. In our cross-sectional sample, we observed a) progressive reduction of neuromelanin content in the substantia nigra pars compacta detected with NM-MRI (average maps from 25 HC, 42 RBD and 10 PD) and b) progressive reduction of the nigrosome-1 signal intensity (dorsal nigral hyperintensity - DNH - or 'swallow-tail sign') in SWI images (average maps from 32 HC, 44 RBD and 102 PD) from HC to RBD to manifest Parkinson's.

- Abstract:

- It includes a reference to AD patients even though they are not included in the section describing participants. Please remove or clarify whether PD were directly compared to AD. Otherwise, interpret results with caution.

We agree with the reviewer that the reference to AD in the abstract is unclear, and we have removed it. In a separate research paper (Rolinski et al., 2015) we compared PD and HC from OPDC with an external sample of AD from Zamboni et al., 2013, acquired with the same scanner and protocol. The results showed that the reduction in BGN connectivity was present in PD but not AD, suggesting that the reduction was disease-specific and not related to neurodegeneration in general. We have better clarified this in the "Findings to date" section and removed the reference to AD from the abstract. The relevant text now reads:

Abstract: *"Brain function, as assayed with resting fMRI yielded more substantial differences, with basal ganglia connectivity reduced in both early Parkinson's and RBD. Imaging of the substantia nigra ..."*

(Page 14) *"When comparing HC and iPD from the OPDC-MRI sub-study with a group of patients with Alzheimer's acquired with the same scanner and protocol (Zamboni et al., 2013), no BGNFC alteration was found in Alzheimer's, providing some confidence that the effect we observed in Parkinson's is pathology-specific."*

- Participants:

- Collection of MRI data started in 2015 and follow-up in 2015. It seems that by now there would be sufficient clinical data collected in these groups to run an analysis that looks at whether baseline imaging metrics predict the progression on clinical measures. Please include a description of how many follow-up scans have been collected so far (perhaps include in a table or modify an existing table).

Since the aim of this paper was to describe the rationale for a cohort's creation, its methods, baseline data and its future plans, new results on follow-up data go beyond the scope of this cohort paper, and will be the scope of future research papers. However, we agree with the reviewer that is useful to include in this paper what is the current amount of follow-up data, although acquisition is still ongoing. We therefore included this information at the end of the "Long-term follow-up" section.

“...In this way we hope to assess their trajectory during the prodromal phase and capture the associated brain changes. So far (December 2019) we have collected 91 follow-up scans: 47 iPD, 2 PD-LRRK2, 19 RBD, 2 RBD-GBA, 21 HC.”

- The paper would have benefited from preliminary analyses of the baseline imaging data vs. baseline clinical data (UPDRS/MOCA).

As the reviewer correctly points out, analyses looking at baseline clinical in relation to imaging data are one of the ways in which our data can be used to infer on PD pathology and progression. Some of our previous research papers have tested the association of imaging data and clinical data.

Regarding associations between imaging and motor performance, in Szewczyk- Krolkowski et al., 2014 we did not find any significant correlation between average BGN connectivity and UPDRS-III, Hoehn & Yahr, disease duration, or age (neither in the PD nor the control groups). Similarly, in Rolinski et al 2015, the BGN connectivity measures extracted from rfMRI were significantly different between PD patients and controls, but did not correlate with UPDRS or disease duration. In Rolinski et al., 2016 we also compared the parameter estimates extracted from the BGN within the areas showing between-group difference to control for laterality. In Parkinson’s disease subjects we compared (i) those with unilateral versus bilateral signs on the UPDRS III; and (ii) those with a higher UPDRS III scores for the left side with Parkinson’s disease subjects with higher UPDRS III scores for the right side. No significant differences were found in either case. To further investigate the influence of laterality of symptoms with functional connectivity we correlated the parameter estimates extracted from the BGN with the contralateral UPDRS III score. No significant correlation was found.

Also when looking at susceptibility-weighted MRI of the substantia nigra we did not find a link between imaging and UPDRS-III. In particular, amongst RBD patients, there was no difference in age or UPDRS III score between those with dorsal nigral hyperintensity (DNH) present and those with DNH absent, despite the fact that the latter had significantly lower dopaminergic SPECT/CT signal in the putamen. The mean DNH intensity also did not correlate with UPDRS III scores in either RBD patients or PD patients.

Regarding associations with cognition, in Klein et al., 2018 we used cortical measurements of macro- and microstructure and performing multi-modal linked ICA on structural, quantitative T1 and dMRI in early Parkinson’s. We found significant differences between Parkinson’s and controls in a fronto-parietal component primarily driven by altered cortical diffusion. These were related to cognitive performance (Montreal Cognitive Assessment – MoCA).

We have now incorporated details about the tested associations in the main text:

(Page 14) “We found significant differences between Parkinson’s and controls in a fronto-parietal component primarily driven by altered cortical diffusion (FA and MD). These were related to cognitive performance (Montreal Cognitive Assessment – MoCA).”

(Page 14) “We did not find significant association between BGFC and motor performance (UPDRS-III)(Szewczyk- Krolkowski et al., 2014; Rolinski et al 2015) also when taking laterality into account (Rolinski et al., 2016).”

(Page 15) “SWI images showed progressive reduction of the nigrosome-1 signal intensity (dorsal nigral hyperintensity - DNH - or ‘swallow-tail sign’) from HC to RBD to manifest Parkinson’s in our cross-sectional sample (supplementary figure S2). In our RBD imaging cohort 27.5% of patients have pathological nigrosome imaging, defined as absence of the DNH, compared with 7.7% of controls and 96% of patients with Parkinson’s disease. Intriguingly, RBD patients with DNH absent did not have lower UPDRS-III scores than those with DNH present, but did have reduced dopamine transporter binding in the striatum, suggesting that nigral SWI may be able to identify individuals with dopaminergic decline (Barber et al., 2020).”

- Can you comment on the stability of the diagnosis at follow-up? Any statistics on drop-outs or exclusion because of conversion to atypical forms of parkinsonism? Any other neurological disorder besides PD or any concurrent movement disorder such as dystonia?

Following the reviewer’s suggestion, we have now added information about the consistency of diagnosis at clinical follow-up and indicated the number of RBD participants who converted after receiving baseline MRI. Future work will specifically look at the relationship between baseline imaging and clinical follow-up.

The relevant text now reads (Page 15):

“At the latest clinical follow-up visit 2 PD participants have been re-diagnosed with multiple system atrophy (MSA), 3 with progressive supranuclear palsy (PSP) and 1 with motor neurone disease. For the remaining PD patients the diagnosis was confirmed with diagnostic confidence over 90%, except one with 80% and one with 55%. Among RBD participants, 11 converted after receiving baseline MRI: 6 converted to PD, 2 to dementia with Lewy bodies, 2 to MSA and 1 to pure autonomic failure.”

- Were there any behavioral measures collected in addition to the gold standard UPDRS? Any planned analyses?

Thank you for this comment, which was also raised by Reviewer 1. Assessments of patients were via self-completed questionnaires and from research visits, using standardised and validated scales. Due to word limit constraints, we decided to add a supplementary table with these details and refer to it and the relevant publications in the main manuscript. However, if the reviewers and the editor think this should be moved into the main text, we would be more than happy to move the table.

The text now reads (Page 7):

“Participants receive extensive assessment in designated research clinics as part of their participation in the OPDC Discovery cohort. The assessment, performed by a nurse and neurologist, includes a structured general medical interview, detailed characterisation of motor and non-motor features, and cognitive assessment (see Supplementary table S1, Szewczyk-Krolikowski et al., 2013 and Rolinski et al., 2015 for details).”

- Resting-state fMRI:
 - Please clarify how the data were collected: eyes open (fixation cross/no fixation) vs. eyes closed.

We have now added this detail in the text, since it was only present in Table 1. The text now reads:

(Page 10) *“...particular interest for the OPDC-MRI cohort. rfMRI data were acquired with eyes open. In 28 iPD patients we also repeated the rfMRI sequence ...”*

- I agree with the authors that resting state fMRI is not affected by the subject's performance and this makes the technique optimal for patients who may have cognitive problems and struggle with task instructions. However, it is well known that in PD this technique gives variable results with many previous studies struggling to validate the results. A major issue here is the fact that a key symptom in PD is tremor at rest (sometimes unilateral, sometimes bilateral) and this likely contaminates the activity. Do the authors control for that in any way? EMG activity during resting state? Rest tremor score from UPDRS as a covariate in the analysis? If not, this should be addressed in the discussion.

We agree with the reviewer that movement artefacts are a major problem in rfMRI, and especially in this population. In order to take this issue into account we adopted several strategies. First, we excluded patients who had more than mild head tremor or presence of dyskinesia/dystonia. Second, we tried to minimise head movement in the scanner by using padding and foam cushions during acquisition. During the analyses mean relative displacement was calculated during motion correction with MCFLIRT to exclude subjects with excessive motion and the impact of motion parameter time series was removed from the data together with the contribution of artefactual ICA components. Finally, when testing differences across groups, we verified that there was no significant difference in mean relative displacement among groups (*Rolinski et al., 2015; Rolinski et al., 2016*).

We have now rephrased some text to include these details, also in response to reviewer 1

(Page 6) “Parkinson’s patients with more than mild head tremor or presence of dyskinesia/dystonia were excluded since movements artefacts would be likely to be too severe to obtain usable images.”

(Page 7) “Scanning was performed at the Oxford Centre for Clinical Magnetic Resonance Research (OCMR) using a 3T Siemens Trio MRI scanner (Siemens, Erlangen, Germany) equipped with a 12-channel receive-only head coil and foam cushions were used to minimize head motion. The neuroimaging protocol includes both structural and functional sequences and lasts approximately 45-50 minutes.

(Page 10) “Mean relative displacement was also calculated to potentially exclude subjects with excessive motion and as possible confound metric in further analyses (e.g. to verify that there was no significant difference in mean displacement across groups (Rolinski et al., 2015,2016)). [...]. The contributions of the motion parameter time series and of artefactual components were regressed out from the data.”

- The authors used an ICA approach implemented in FSL along with a dual regression to analyze the data. The supplementary material contains an example of the mean resting state activity within the basal ganglia network (defined in MELODIC). It is not clear if the statistical map is that of PD or controls. It is mentioned that basal ganglia functional connectivity was reduced in early PD. Connectivity with what region? It would help to plot that (results of dual regression analysis) rather than the BG network alone.

- P value? Correction for multiple comparisons. Please include details on this especially when describing figures (e.g. BG network).

We apologise for the lack of clarity regarding supplementary figure S1 and relative legend. We have now regenerated the figure showing the BGN template together with the group differences across HC, RBD and PD (referring to this analysis in the text). We also added details about p-values in the colour bar and in the legend.

Figure S1. Resting state fMRI. Yellow underlay: basal ganglia network (BGN) template map, part of the template created with group ICA from 45 healthy controls external to the OPDC MRI sub-study (threshold of $z=5$ is applied for display purposes only; more details in Griffanti et al., 2016, and full template available online). This was used in dual regression to extract subject-specific maps, which were then compared across groups. Results of the one-way ANOVA across 3 groups (n. PD = 103, n. RBD = 67, n. HC = 68) with post-hoc t-tests showed reduced functional connectivity in Parkinson's compared to controls (panel a, red) and reduced functional connectivity in RBD compared to controls (panel b, blue). Values displayed are $1-p > 0.95$, corresponding to $p < 0.05$, corrected for multiple comparisons using threshold-free cluster enhancement. The bottom panel shows an overlap of the results. Full maps are available on NeuroVault (<https://identifiers.org/neurovault.collection:5686>). This is a replication of the analysis performed in Rolinski et al., 2016 on a smaller sample (PD = 48, n. RBD = 26, n. HC = 23). These results are in line with the original study, although the effect size is smaller.

(Page 14) “Our measure of BGFC was reproducible across different analysis settings (Griffanti et al., 2016); however, these group differences have diminished with increasing sample size (supplementary Figure S2)...”

VERSION 2 – REVIEW

REVIEWER	Roxana Burciu University of Delaware
REVIEW RETURNED	03-Jun-2020
GENERAL COMMENTS	I believe the manuscript is ready for publication.